

# A robust gap-filling method for Net Ecosystem Exchange based on Cahn-Hilliard inpainting

Yufeng He[1], Mark Rayment[1]

[1]School of Environment, Natural Resources and Geography, Bangor University, UK

*Correspondence to*: Yufeng He (afp23e@bangor.ac.uk)

**Abstract.** Traditional gap-filling approaches adopt a temporally linear perspective on data; whether synthesizing data statistically within a moving window, or using complex functions based on a "best-guess" understanding of the processes driving exchange. The former approach is limited in its ability to capture non-linear trends, and the latter is limited in situations where the flux response to driving variables is poorly understood or unknown (e.g. the response of gas exchange to

water table depth in wetlands). Rearranging time-averaged half-hourly net ecosystem exchange (NEE) into a 48*N matrix has been used to visualize NEE as a "flux fingerprint" and suggests a different way of filling data gaps. In this paper, we introduce an image processing technique known as image inpainting to fill gaps in this two-dimensional representation of a one-dimensional data. This has the advantage that any short-term structure can be accommodated without expressly implying any particular functional response to driving environmental variables, and medium-term temporal structure (i.e. day-to-day

covariance) can be incorporated into gaps in the flux signal. In this way, data gaps are filled solely using information contained in robust, primary data. This new method compares favorably with the marginal distribution sampling (MDS), when tested on twelve European-Flux datasets with four types of artificial gaps. Furthermore, we show that how random structures or noise embedded in the signal affect the gap-filling performance, which can simply be improved through a de-noising procedure by using a Fourier transform algorithm. The inpainting-based gap-filling approach is more effective than

MDS on the de-noised data.

## 1 Introduction

The eddy covariance (EC) technique used for measuring the fluxes of greenhouse gases (GHG) and energy has flourished over the past 25 years (Baldocchi, 2014). It is considered as the only method that provides a direct sense of the gas/energy exchange at the biosphere–atmosphere interface at the canopy scale (Baldocchi, 2003; Baldocchi et al., 1996). Globally,

more than 400 sites are equipped with gas sensors with high temporal resolution monitoring gas exchange and dozens of groups have produced time-series spanning years and decades (Baldocchi, 2014). While the expansion in use of EC has greatly helped us in the understanding of land-atmosphere exchanges, the method does not yet provide perfectly reliable data on the magnitude and location of GHG sinks/sources as the result of several theoretical and practical limitations. Notably, the method is intrinsically limited to use in generally flat terrain with generally uniform vegetation and an adequate footprint

area (Baldocchi, 2003; IPCC, 2000); such limitations are unassailable. Beyond this, however, data are lost by data rejection



when theoretical requirements are not met, e.g. during low-turbulence periods, by other data-quality controls, or often by partial or complete equipment failure (Aubinet et al., 1999; Foken and Leclerc, 2004; Goulden et al., 1996; Papale et al., 2006). Such gaps can account for 20-60% of an annual dataset of the net ecosystem exchange (NEE) (Falge et al., 2001; Moffat et al., 2007). Thus, in spite of theoretical limitations, dataset incompleteness is a major hindrance to the impact of EC

in the widespread quantification of GHG exchange.

Despite their incompleteness, fragmented data sets may contain sufficient information for short-term (i.e. half-hourly) interpolation and for limited evaluation of process-based models, however intactness is a fundamental requirement for estimating annual carbon budgets and for comparison with other biometric measurements. Traditional approaches to tackling gap filling in NEE measurements are mainly based on the idea of correlating the flux with other driving environmental

variables (e.g., temperature, global radiation, water vapour, etc.) where fewer gaps and more predictable (or at least, more well understood) temporal variation occurs. This has led to a fruitful development of gap-filling techniques, broadly classifiable into three categories: non-linear regression, moving window average and artificial neural network (ANN). A comprehensive comparison (Moffat et al., 2007) of fifteen such methods based on 10 benchmark datasets showed that different techniques performed almost equally well, with ANN slightly (but not significantly) better because it is better able

to replicate underlying patterns in the data. The reason why 15 independent methods resulted in similar performance however, remained unexplained. A plausible explanation is that the gap-filling efficiency was ultimately limited by the noise in the signal. Here "noise" represents stochastic, unstructured variation, unrelated to known environmental drivers (see details below).  As noise becomes larger relative to the "real" signal, it becomes harder for any gap-filling algorithm to distinguish the real information that needs to be replicated. Thus, irrespective of gap-filling method, the estimation variance

may be primarily a reflection of the variance in signal noise rather than the efficiency of the method itself.

Introducing auxiliary information from secondary environmental variables can assist in re-construction of the flux time-series but this is limited in two situations. Firstly, the flux response to driving variables may be poorly understood or unknown (e.g. the response of gas exchange to, for example, water table depth in wetlands), not least because of any non-linearity of the system (i.e. simple regression functions are not capable of capturing all variations in the system) (Lasslop et

al., 2010). The form and parameters of regressions and look-up tables are site-specific, hindering progress in standardising the estimation of carbon exchange and reducing biases among sites (Reichstein et al., 2005). Secondly, any uncertainties or errors in environmental variables that are used in regressions or to train ANN propagate into the final NEE estimation. A further limitation in using ANN is that their intricately integrated structure makes it difficult to track the effect of, and noise introduced by, input variables, some of which may have limited predictive power and may even be redundant in terms of

contributing to the real signal (Tu, 1996).

Rearranging a half-hourly time-series of NEE into a 48 * N matrix (where the rows represent the time of day (i.e. 48 half-hourly periods) and the N columns represent the day of year) provides us a way of visualising the time-series in two dimensions, commonly known as the flux fingerprint figure. In this paper, we present a gap-filling method of NEE based on a technique known as image inpainting (Bertalmio et al., 2000) which has become mature in fixing corrupted 2-dimensional



images but not been used in tackling the gap-filling in time-series. This method has the advantage that any temporal structure (e.g. daily and half-hourly covariance) is better incorporated into gaps in the flux signal without implying any particular functional response to driving environmental variables. Similar to the principle behind ANN-based machine learning, the image inpainting technique can sense any underlying structure in the time-series by iteratively and smoothly propagating

information (see the methodology section for details). Moreover, compared with traditional methods such as the two standardized ones adopted by Carboeurope and FLUXNET (Papale et al., 2006) where many inputs (e.g. temperature, radiation, u*, VPD, etc.) and complex functions are needed, data gaps are filled solely using information contained in the flux data themselves, largely simplifying the gap-filling process and avoiding potential uncertainties introduced by auxiliary information (e.g. poor quality of auxiliary information and over-fitting).

## 10   2 Materials and methods

### 2.1 Data description

Twelve (12) years of data from 6 European sites are selected for conducting the comparisons of the gap-filling performance. The Level-3 NEE products (see code and data availability) are used for implementing the inpainting-based gap filling and the required driving environmental variables are added to run the Marginal Distribution Sampling (MDS) gap filling

procedure. In this study, datasets from the process of quality control (QC) (e.g. u* criterion, spike detection, Steady state tests) (Foken et al., 2004) are used for simulation. The gap percentage varies among sites and years, from 29.5% up to 56.7% (see Table 1).

### 2.2 Gap filling methods and artificial gap type

MDS – Marginal Distribution Sampling (MDS) is a moving window average method where both NEE and the driving

environmental variables are required as inputs for the algorithm (Reichstein et al., 2005). The R Package called REddyProc (Reichstein and Moffat, 2015) is used to implement MDS.

IIP – Image Inpainting (IIP), a fourth-order partial differential equation called the Cahn–Hilliard Equation is solved numerically to propagate information smoothly from outside the data-missing region into it (Burger et al., 2009; Schönlieb, 2015). The inpainted image can be considered as a highly smoothness estimator of the original image and the "smoothness"

was solved gradually until a stable state is reached. A simple example is given below (Fig.1), showing the reconstruction of corrupted images using IIP. Similarly, the fingerprint figure of NEE is converted to a grayscale image and the gaps are then filled by the inpainting algorithm based on the code from the MATLAB Central File Exchange (Schönlieb, 2011).

In order to evaluate the performance of the gap-filling methods on the data points where real values exist, short and long artificial gaps amounting to about 10% of each dataset are considered in the simulations. Concretely, for the short type, half-

hourly gaps are added uniformly randomly to the original NEE signal, while gaps with length of 3-day, 5-day and 7-day are added respectively (Fig 2).





### 2.3 Noise reduction

To start with, we need to clarify what "noise" means in the context here. For a given signal, it can be partitioned into two parts: the trend part and the stochastic part. The trend part is called the de-noised signal and the stochastic part is referred as the noise. The noise characterized the randomness of a signal. As there is no general rule for reducing the noise from a NEE

signal, the following assumptions are made for validating a de-noise method:

1.    Noise has zero mean and symmetric/unbiased distribution;

2.    Covariance between the noise and the de-noised signal is negligible (close to zero);

3.    The difference between the before and after de-noising are small in the cumulative temperature and NEE.

The point 1 and 2 are used to show that the noise part was similar to a stochastic, unstructured and non-correlated signal.

Since the underlying pattern of NEE is unknown, the cumulative and average temperature and NEE are used to show that the important information still remains after the de-noising process (see details in Results).

A simple method based on the Fourier transform of an entire time-series is used to reduce noise in the NEE and temperature signals. This process is illustrated by the block diagram:

$$x(n) \rightarrow \boxed{\mathcal{F}\big(x(n)\big)} \rightarrow \boxed{Threshold: \hat{x}(k) = g \cdot \mathcal{F}(x)} \rightarrow \boxed{\mathcal{F}^{-1}\big(\hat{x}(k)\big)} \rightarrow y(n)$$

where x($n$) is the original "noisy" signal in the time domain ($n$), with any gaps initialized with the mean value of the rest of the signal. F and F $^{-1}$ are the fast Fourier transform and its inverse respectively. $\hat{x}(k)$ is the filtered signal at frequency (k) and y($n$) stands for the de-noised signal. The threshold step was carried out using a simple binary function:

$$g(a) = \left\{ \begin{array}{l} 0, \ |a| \leq T \\ 1, \ |a| > T \end{array} \right. ,$$    (1)

Two more sophisticated noise reduction techniques, the short-time Fourier transform and wavelets (each using various sized

windows) were also tested in our study, but did not show distinct advantages over the simple Fourier transform, and the results are not presented here.

A dimensionless quantity is used to measure how much noise has been removed by the de-noising process. In image processing, the quality of a signal can be expressed quantitatively as the signal-to-noise ratio (SNR) (Schowengerdt, 2006), denoted as:

$$SNR = \frac{\sigma_{signal}}{\sigma_{noise}} ,$$    (2)

where $\sigma_{signal}$ and $\sigma_{noise}$ are the standard deviation of post-filter signal and the standard deviation of the filter-out signal (noise) amplitude in the Fourier domain, respectively.

### 2.4 Analysis

The two gap-filling methods are applied to the original and the noise-reduced NEE data from 12 years of measurements at 6

European sites respectively. Following Moffat et al. (2007), we assume that the differences between the traditional methods are negligible, and therefore comparisons were only conducted between IIP and MDS. Initially, we apply the two methods to



the original NEE datasets and measure their performance on four types of artificial gaps, including short random gaps and long gaps up to 7 days. Further simulations are then conducted to show how noise or random structures in the signal may affect the gap-filling performance by partitioning the original signal using Fourier transform.

## 3 Results

### 3.1 Gap filling the NEE data with artificial gaps

Figure 3 shows an example of the comparison of the gap-filling performance between IIP and MDS on the post-QC NEE data with artificial gaps at site DEGri 2012. The temporal patterns revealed by the two gap-filling methods are very similar across all four gap types. Clear diurnal and seasonal variations are well captured by both methods. In contrast to MDS, contour structures and boundaries generated by IIP are smoother and less noisy. This can be seen more clearly from the large gap (~ 2 weeks) in the middle of the year. Effects of the gap type on the gap-filling performance are minimum and can hardly be noticed from the contour plots, which may suggest that IIP is able to reconstruct the signal even with the occurrence of the long-gap types up to 7 days.

The difference of gap-filling performance between IIP and MDS is further evaluated for all twelve datasets at the data points of artificial gaps where the true NEE values are available. The gap filling error is simply calculated by taking the difference of the estimated and real values at those data points. The error distributions represented by the error bar plot (Fig. 4) again confirm what we have seen previously from the contour plots, showing little difference between the two methods, i.e. comparable means and variances in the gap-filling error. Mean values close to zero suggests that both methods provide nearly unbiased estimations for the NEE signal. Combining all twelve datasets categorized by gap types, we see comparable statistics between the two methods (Table 2) and more importantly, no significant dependence of the gap-filling performance on the gap type is found for both methods.

In terms of the goodness of estimation represented by one standard deviation (i.e. the span of error bar), at site ITRo3 2013 the gap-filling error shows the most variation, while it has the least variation at site UKAMo 2010 for the four gap-type scenarios. This raises a question that where the variations among datasets originate from. In other words, why are the estimations from some datasets always better than the others? The span of error bar should be nearly zero for an ideally clean, noise-free image. As noise increases, the span becomes wider. We will address this problem in the next part by assuming the variation of the gap-filling error originates from the random structures in a signal.

### 3.2 Random structures/noise affect the gap-filling performance

To start with, the temperature and the NEE signals are patitioned into two components respectively. To check that the de-noising procedure does not introduce bias, the average and cumulative temperature and NEE are compared before and after the signal de-noising process. An example of the de-noising is shown in Fig 5 for site UKAMo 2010. The average and cumulative temperature signals are almost identical (Fig.5a), demonstrating that the system energetics remain the same even





though some structures of randomness has been removed from the time series of temperature. Furthermore, the distribution of the removed part of temperature or simply the noise distribution shows a good agreement with the normal distribution (Fig. 5c), implying a Gaussian-structured noise embedded in the original temperature signal.

A similar result can be found in the de-noised NEE (Fig. 5b) even though the SNR (~1.2) is much lower than that of

temperature, suggesting that the NEE is initially noisier than the temperature. The value of SNR is determined by the thresholding step (Eq. 1) and for de-noising the NEE signal, 1.2 of SNR was found to be approximately a lower limit of the noise removal in order to maintain a clear diurnal and seasonal variation (Fig. 5b bottom-right). We show this largely smoothed NEE to demonstrate that the average and cumulative NEE after the de-noising are still good approximates to the original ones and for any less smoothed NEE with higher SNR values (see Fig 6) the cumulative NEE fits even better.

Unrealistic fluctuations of the original NEE appear mostly at night-time and the de-noising method seems to fix this, as a traditional regression method would work, by replacing the night-time NEE with some simple variations (Fig. 5b top-left), which is also the main cause for the deviation of the accumulative de-noised NEE from the original (Fig. 5b bottom-left). Intuitively this abnormality in the NEE at night-time supports our speculation that noise exist in the NEE signal, which would affect the gap-filling performance by introducing the error variations. The distribution of the noise part of NEE,

however, is not a good fit to the normal distribution but a rather steep, symmetry shape around 0, which is more like to be a t-distribution (Fig. 5d). This may suggest that the noise type embedded in the NEE, as shall be expected, is more complicated and different from a normal one found in the temperature. The covariance between the noise part and the de-noised part are very close to zero ($\sim 10^{-16}$) for both the temperature and NEE. A further investigation on the statistical feature of the noise is beyond the discussion of this study, however, this result has already provided an evidence that the noise parts removed from

the original signals have some nice statistical feature which satisfies the criteria as proposed previously, i.e. it had zero means, symmetric distributions and negligible correlations/covariance with the real signal. Moreover, both the temperature and NEE remain almost unchanged in their average and cumulative quantities, enhancing the robustness and validity of this de-noise method.

Two sites, the UKAMo 2010 and ITRo3 2013 (see Table 1) are selected, as two extreme examples of the gap-filling error

variation, for a further investigation on how the noise embedded in the signal affects the gap-filling performance as shown in Fig.4 where the gap-filling error is found to be various among sites. We adjust the threshold (Eq. 1) so that an increasing amount of noise is gradually removed from the original signal until a highly smoothed one is reached (Fig. 5b). Only the cases with the random artificial gap type are presented here as it has been shown from the previous part that the gap type has little impact on the performance. Four of the de-noising states for the two sites are shown in Figs. 6 & 7 respectively. The

NEE fingerprints become smoother as the SNR value decreases (i.e. more structures are removed from the original signal), with more visible periodic variations showing up. Because the inpainting method is a high-order PDE algorithm that pursues the smooth solutions, it produces a near-perfect reconstruction of the NEE images when the data are highly de-noised, clearly outweighing MDS (see Fig. 6d & 7d). Noticing that the starting value of SNR for ITRo3 2013 is significantly larger than that for UKAMo 2010; it seems that more noise needs to be removed from ITRo3 2013 to reach a similar level of



smoothness as UKAMo 2010. This further suggests that the larger variation of the gap-filling error found in ITRo3 2013 comes from an initially higher noise level embedded in the signal, which supports our speculation that the gap-filling performance is largely affected by the noise within a signal (Fig. 3). Because of the existence of these random structures, the gap-filling performance was found to be so similar between the two gap-filling methods (Fig. 4).

The response of the gap-filling error to the value of SNR can be seen most clearly in Fig. 8. Although the error decreases as expected for both methods as the SNR value decreases, the improvement gained from IIP is faster and better than that from MDS. Moreover, the uncertainty of estimation (i.e. error bar in Fig. 8) for IIP shows a trend to converge to zero as what an ideal performance should be for a noise-free image, compared with a converged constant significantly larger than zero for MDS. Although small in magnitude, this flux error could be accumulated and propagated into a large one in an annual flux

estimate. To summarize the two gap-filling methods, MDS, being based on a moving-window average algorithm, is a lower-order approximation to a time series, while IIP, being based on a higher-order non-linear equation, can sense and integrate more information in the process of gap-filling.

## 4 Discussions and conclusions

We have seen that the inpainting-based gap-filling method (IIP) is highly comparable with the widely-used Marginal

Distribution Sampling (MDS) in all simulation cases proposed in this study (Fig.4 & Table 2). Evidence from the blurriness in the finger-print figures of NEE (e.g. Fig. 3), the unrealistic patterns at night-time (Fig. 5) and the gap-filling error variations (Fig. 4) pointed out the existence of noise in the signal, therefore we speculated that some random structures in the signal affect the gap-filling performance and contribute to the error variation. When the NEE data were de-noised by the simple Fourier transform, though both methods showed better accuracy of estimation, IIP was more effective in terms of

capturing the smoothness (Fig. 8).

Nevertheless, a natural and fundamental question is, did we discard noise or real signal through the de-nosing process? To our knowledge, there is no definite answers to this question because the noise and signal are not distinguishable unless we know precisely and ahead of time what we are looking at. In contrast to de-noising an image or searching for objects from echo soundings, for instance, establishing criteria for identifying a noise-free NEE signal is currently impossible because we

do not, in fact, understand the underlying, process-based structure of NEE clearly. Our simplified de-noise algorithm based on the Fourier transform was capable of extracting the dominant variations in the signal while maintaining the average and cumulative quantities. Moreover, the removed parts from the original signals, or the so-called noise here, showed some good statistical features (Fig. 5), i.e. zero-mean, unstructured and non-correlated random structures. It is not, however, sufficient for providing a general rule for de-noising NEE (or other driving variables) because the performance of de-noising procedure

depends on the amount of real signal underlying the time-series and the criteria used to distinguish real signal from noise. Taking the most conservative view (i.e. assuming that the NEE data are noise free after quality control), IIP, MDS and even other gap-filling methods are nearly equally good (Moffat et al., 2007). This suggests that these seemingly independent





methods are simply alternative information processing machines, achieving the same level of approximation of a time series. In turn, this raises the prospect of unifying these gap-filling methods and adopting the most parsimonious.

In inpainting, only the target signal is needed to drive the gap-filling process. This simplicity is a distinct advantage of IIP because its internal coherence prevents potential biases being introduced from errors/incomplete auxiliary data and/or best-

guess functions relating auxiliary data to NEE. Similarly robust methods are also found in other signal-processing techniques used to reconstruct noisy signals, i.e. singular spectrum analysis (Buttlar et al., 2014) and the discrete cosine transform (Garcia, 2010), however the utility of these methods for gap-filling NEE datasets remain untested.

While IIP shows some clear advantages over the traditional methods, some noteworthy limitations of this method need to be indicated. Firstly, IIP performed less well for long gaps where the information density is low (i.e. diameters of the gaps are

big). This is especially true where extrapolation into long gaps at the beginning or the end of a time series is needed. In analogous situations where IIP is used to reconstruct missing areas in images, techniques based on finding and copying similar texture structure from other patches can be further explored (Bertalmio et al., 2003). Applying a similar approach to gap-filling NEE would require a hybrid of IIP for texture rendering and process-based understanding ecosystem dynamics (Knorr and Kattge, 2005) for texture mapping. Secondly, IIP is a purely numerical algorithm and cannot yet explain any

system function. Notwithstanding this, however, the accuracy of IIP as an unsupervised process for filling artificial gaps, particularly when coupled with a de-noising algorithm, may contribute to bringing into focus underlying ecological and meteorological mechanisms not identifiable *a priori*.

In this paper, we show that the image inpainting (IIP) is a simple, compact and robust approach for gap-filling NEE, that it performs at least as well as a more complex gap-filling method, and we conclude that IIP should be added to the group of

gap-filling methods for further research on gap-filling NEE. Evidence has been shown that the signal noise ultimately limits the gap-filling accuracy and de-noising the signal before the gap-filling procedure improves accuracy of estimation without introducing bias.

**Code and data availability**

The R Package called REddyProc for implementing MDS can be obtained either from R-Forge (https://r-forge.r-

project.org/projects/reddyproc/) or the CRAN repository. The MATLAB code for implementing image inpainting is available from the MATLAB Central File Exchange (http://uk.mathworks.com/matlabcentral/fileexchange/34356-higher-order-total-variation-inpainting). In particular, the M-file called bvnegh_inpainting_convs.m was used for the IIP-based gap filling.

All datasets used in this paper can be directly requested from the European Fluxes Database Cluster (http://www.europe-

fluxdata.eu/). Please also refers to the details of data information from the request.





**Acknowledgement**

This work is supported by the joint PhD program between Bangor university – China Scholarship Council. We greatly thank the data providers from the European Fluxes Database Cluster for allowing us use their data in this study. We also thank Dr James Gibbons for his valuable comments on the manuscript.

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





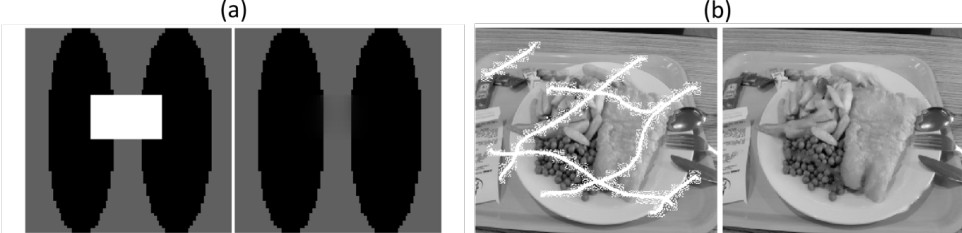

Figure 1. Examples of the Image Inpainting. (a) A simple structure with artificial square gaps in white area (left) and its reconstruction (right); (b) A real photo corrupted with gaps (i.e. white stripes) (left) and its reconstruction (right).

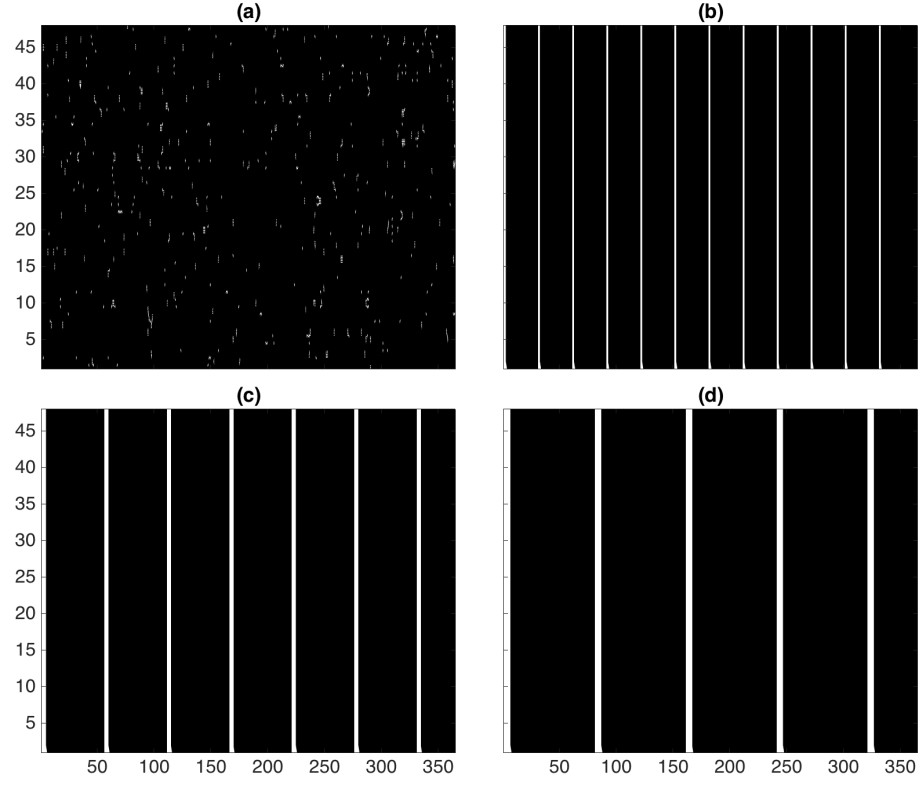

Figure 2 Four gap types generated are (a) random gaps; (b) 3-day; (c) 5-day; (d) 7-day. White strips represent the gap positions. Number of gaps for each gap type was about 10% of the whole year (i.e. ~1752 data points).




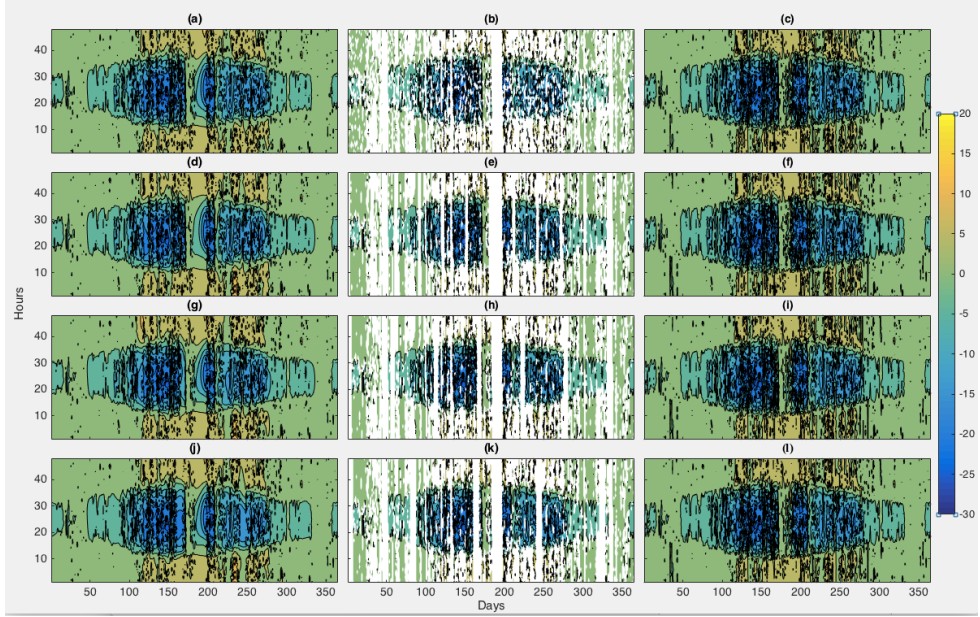

**Figure 3. Image inpainting vs Marginal Distribution Sampling for four gap types at site DEGri 2012. The middle column (i.e. (b), (e), (h) and (k)) are the original NEE data with random, 3-day, 5-day and 7-day artificial gaps respectively. The left (i.e. (a), (d), (g) and (j)) and right (i.e. (c), (f), (i) and (l)) column represent the gap-filling results from IIP and MDS accordingly.**



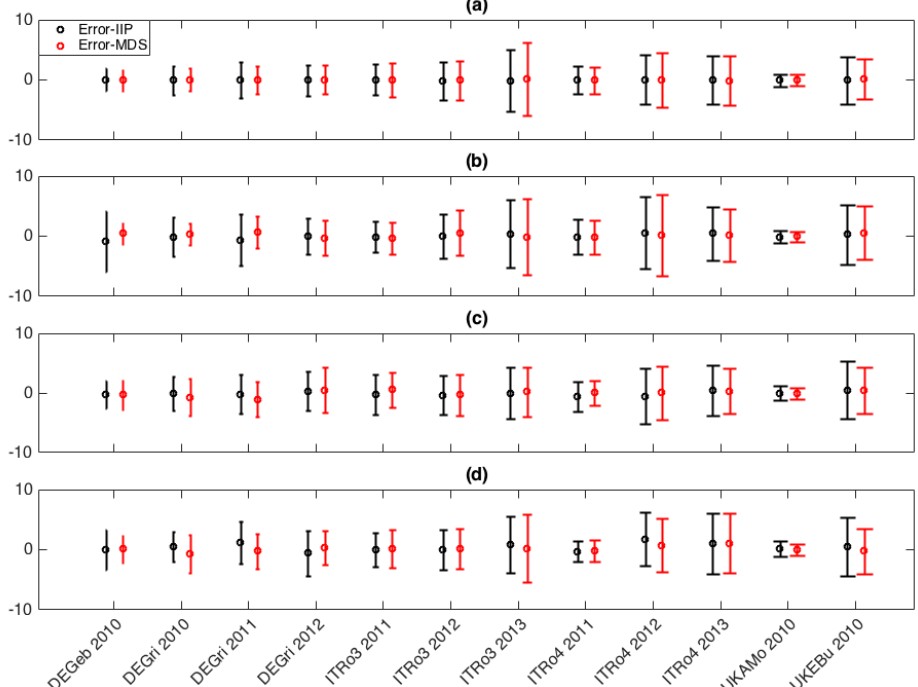

**Figure 4. Comparisons of the gap-filling error between IIP and MDS for the four gap types, i.e. (a) Random gaps; (b)3-day gaps; (c) 5-day gaps; (d) 7-day gaps. Unit of the error values are in μmol m-2 s-1. Error bar plot with mean ± one standard deviation of the absolute errors for each dataset.**





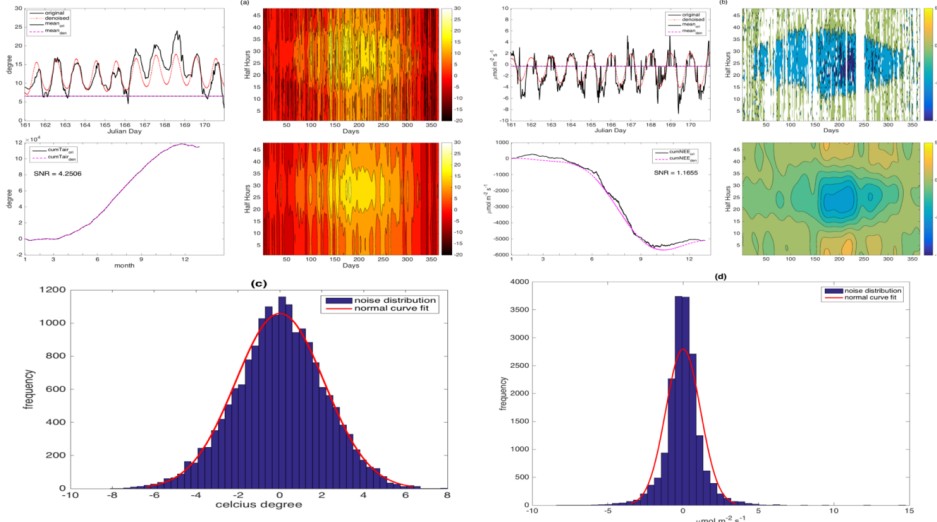

**Figure 5. An example of highly de-noising temperature and NEE at the site UKAMo of year 2010. Raw and de-noised half-hourly (upper-left) and cumulative (lower-left) temperature (a) and NEE (b), together with the 2-d visualizations (fingerprints). Noise distribution of temperature (c) and NEE (d).**



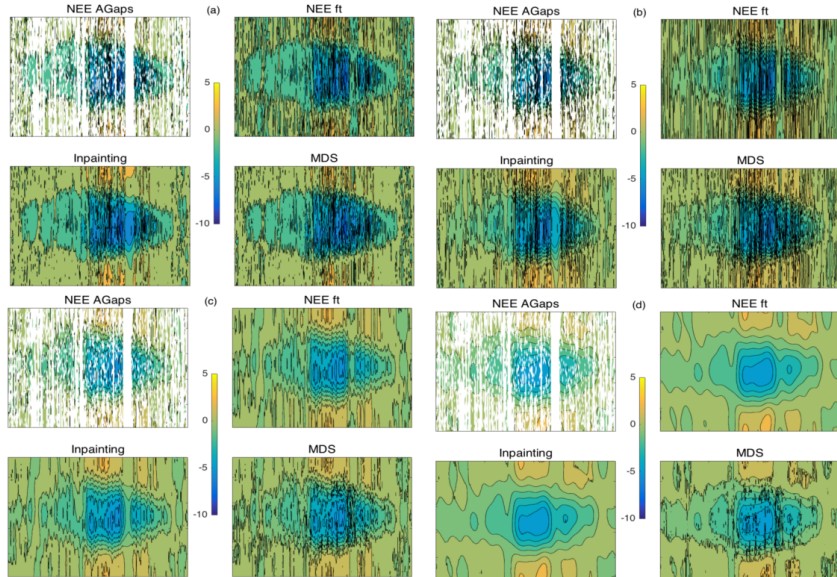

**Figure 6. Data from UKAMo_2010. De-noising the NEE dataset and its gap filling. NEE ft is the de-noised NEE by Fourier transform. NEE AGaps stands for the post-denoise NEE with artificial gaps and real gaps. SNR levels are (a) 2.34, (b) 1.68, (c) 1.30, (d) 1.17.**




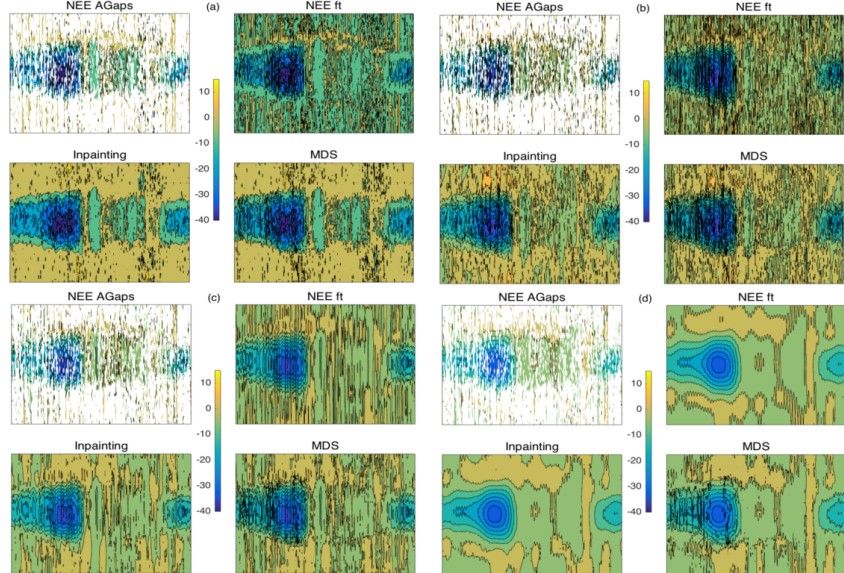

**Figure 7 Site: ITRo3_2013. De-nosing the NEE dataset and its gap filling. NEE ft is the de-noised NEE by Fourier transform. NEE AGaps stands for the post-denoise NEE with artificial gaps and real gaps. SNR levels are (a) 24.87, (b) 1.90, (c) 1.50, (d) 1.35.**



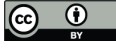

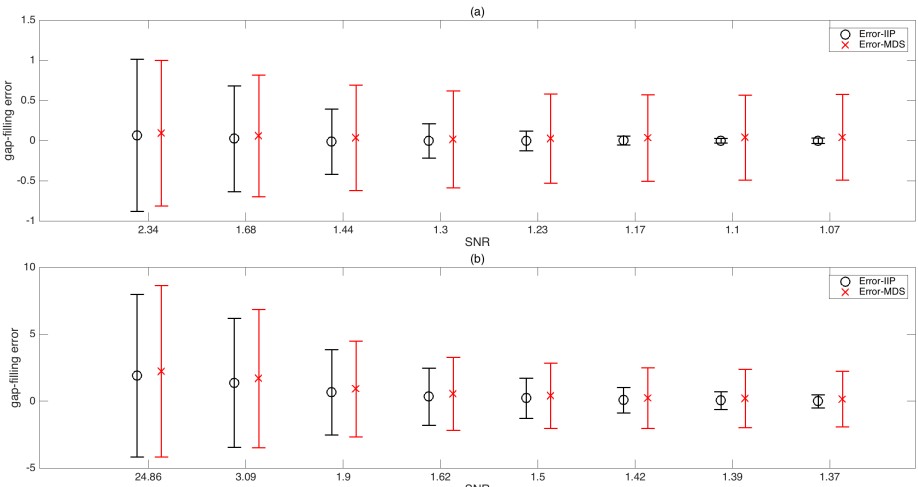

**Figure 8.** Gap-filling errors response to the ratio of energy remaining after de-noising (SNR). (a) Site: UKAMo_2010; (b) Site: ITRo3_2013. The gap-filling error simply refers to the difference between the gap-filling values and the real value at artificial gaps. Error bar plot stands for the mean ± one standard deviation of the errors.

5   **Table 1.** Description of the datasets from the European fluxes database. The gap percentage was calculated by counting the half-hourly data missing of a whole year (i.e. 17520 of data records for 365 days).

| Site-ID | Location | Vegetation Type | Lat/Long | Year | Gap(%) | Post-QC Gap(%) | PI |
|---|---|---|---|---|---|---|---|
| **UKAMo** | Scotland | Peatland | -3.23°, 55.79° | 2010 | 9.9% | 30.2% | Marc Sutton |
| **UKEBu** | Scotland | Grassland | -3.20°, 55.86° | 2010 | 26.4% | 44.8% | Marc Sutton |
| **DEGeb** | Germany | Cropland | 10.91°, 51.10° | 2010 | 43.1% | 55.9% | Olaf Kolle, Mathias Herbst |
| **DEGri** | Germany | Grassland | 13.51°, 50.95° | 2010 | 9.9% | 31.1% | Christian Bernhofer |
| | | | | 2011 | 17.9% | 37.2% | |
| | | | | 2012 | 11.0% | 29.5% | |
| **ITRo3** | Italy | Cropland | 11.92°, 42.38° | 2011 | 23.6% | 50.1% | Dario Papale |
| | | | | 2012 | 22.0% | 48.4% | |
| | | | | 2013 | 7.1% | 42.7% | |



| ITRo4 | Italy | Savannah | 11.92°, 42.37° | 2011 | 27.7% | 56.7% | Dario Papale |
|--------|-------|----------|-----------------|------|-------|-------|--------------|
|        |       |          |                 | 2012 | 19.1% | 47.8% |              |
|        |       |          |                 | 2013 | 23.4% | 52.2% |              |

**Table 2 Overall statistics of the gap-filling error for all datasets for the four gap types.**

| Gap types | Random | | 3-Day | | 5-Day | | 7-Day | |
|-----------|--------|--------|--------|--------|--------|--------|--------|--------|
| Gap-filling methods | IIP | MDS | IIP | MDS | IIP | MDS | IIP | MDS |
| Sample size | 11817 | 11817 | 11493 | 11493 | 11322 | 11322 | 10593 | 10593 |
| error mean | -0.0307 | 0.0229 | -0.0067 | 0.1654 | -0.0782 | -0.0378 | 0.3377 | 0.1472 |
| Standard dev. | 3.1455 | 3.1948 | 4.1013 | 3.8099 | 3.4485 | 3.3259 | 3.6577 | 3.5091 |

