# Peer review of "A robust gap-filling method for Net Ecosystem Exchange based on Cahn-Hilliard inpainting"

_Geoscientific Model Development, 2016_

## Short Comment (SC1) · 1 Jun 2016

Very interesting work for the contribution to science

---

## Referee Comment (RC1) · Anonymous Referee #1 · 29 Jun 2016

The paper presents the application of an image reconstruction technique (IIP - Image Inpainting) to fill gaps in eddy covariance timeseries of Net Ecosystem Exchange. The authors compare the method proposed with another generally used technique (MDS – Marginal Distribution Sampling) using artificial gaps and de-noised timeseries from 6 eddy covariance sites and 12 years of data, concluding that the IIP method has similar performances respect to the MDS and it is outweighing MDS in case of de-noised timeseries.

The paper is well written but the analysis is weak, not very innovative and not convincing mainly because:

1) The comparison of the two method is based on artificial gaps of up to 7 days; this

is definitely a period too short to challenge the IIP method. In fact, as also the authors say, the drivers used by the MDS method add noise in the performances and for this reason on short gaps it is expected that methods based only on NEE interpolation will work better (the Mean Diurnal Variation (Falge et al 2001) is also bases only on NEE data and will probably perform well in these conditions). The effect of changed environmental conditions will be probably visible on a longer time interval and for this reason the IIP should be evaluated on gaps of 2-3 weeks.

2) The discussion on the noise on the NEE data is interesting and largely correct. However the conclusions related to the study are somehow expected and not proving the goodness of the IIP method. In fact as the authors assert the IIP is an "highly smoothness estimator" and for this reason is it expected that its performances in the image re-construction will dramatically improve is the image is "smoothed" and simplified. It is also expected that with an "oversimplified" NEE time-series the effect of the short term variability of the meteorological drivers for the MDS can only add noise to the result (the potential short term relation driver-output is broken by a filtering applied to the output; for example the fast pulse effect on respiration due to precipitation or the fast reaction of photosynthesis due to cloudy periods).

3) Some of the results interpretation are subjective and not justified. For example the fact that the reconstruction of the larger gap in summer in the DE-Gri site (figure 3) by the IIP is something to be positively evaluated because smoother and less noisy respect to MDS (page 5) needs to be proved. It is possible that the correct reconstruction is the one from the MDS method. . . only artificial (long) gaps can say which method is more close to the original measured data

Reference: Falge et al. (2001) Gap filling strategies for defensible annual sums of net ecosystem exchange, Agricultural and Forest Meteorology 107, pp. 43–69

---

## Author Comment (AC1) · 6 Jul 2016

We are delighted that the referee found this paper well written and thanks for his/her valuable comments. We understand the major concern raised by the referee and see that it lies in the interpretation of results 3.1, which leads to some confusions in the subsequent analysis where we have tried to gap-fill the de-noised NEE signals. Substantial revisions have been made to the result section of the manuscript, including adding a 14-day scenario (see Fig. 2-4, Table 2), and we have made major corrections to the narrative of results 3.1 (see attached file for a revised version of the manuscript). The detailed responses (with the referee's comments quoted) are as follows:

1) About artificial gap length
**Reviewer's comments**: "The comparison of the two method is based on artificial gaps of up to 7 days; this is definitely a period too short to challenge the IIP method. In fact, as also the authors say, the drivers used by the MDS method add noise in the performances and for this reason on short gaps it is expected that methods based only on NEE interpolation will work better (the Mean Diurnal Variation (Falge et al 2001) is also bases only on NEE data and will probably perform well in these conditions). The effect of changed environmental conditions will be probably visible on a longer time interval and for this reason the IIP should be evaluated on gaps of 2-3 weeks."

**Response**: We agreed on challenging the IIP at a longer gap length and we have included a 14-day scenario in the manuscript. The results still show little difference between IIP and MDS in gap-filling a 14-day scenario (please refer to the attached manuscript for a revised result 3.1 and Fig. 3&4). Despite the reviewer's concerns that a long gap would challenge IIP, this was not the case - 1) Smoothly filling the gaps (by IIP) did not necessarily performed less well in terms of the estimation accuracy; 2) Large gaps did not significantly affect the gap-filling performance for either method. (Falge et al 2001) reported that "With the data at hand, we were unable to answer which methods compared best with the artificially removed data, and under what conditions (day, night, functional group, climatic conditions). The residuals between artificially removed and filled data of various sites, methods and gap percentages could not be distinguished by ANOVA". It seems that both methods and (very likely) other methods largely "failed" to recover the nuances of the original signal. This is exactly the reason that drives us to investigate the effects of noise on gap-filling performance in the subsequent analysis.

2) About de-noising the NEE signal
**Reviewer's comments**: "The discussion on the noise on the NEE data is interesting and largely correct. However the conclusions related to the study are somehow expected and not proving the goodness of the IIP method. In fact as the authors assert the IIP is an "highly smoothness estimator" and for this reason is it expected that its

performances in the image re-construction will dramatically improve is the image is "smoothed" and simplified. It is also expected that with an "oversimplified" NEE time-series the effect of the short term variability of the meteorological drivers for the MDS can only add noise to the result (the potential short term relation driver-output is broken by a filtering applied to the output; for example the fast pulse effect on respiration due to precipitation or the fast reaction of photosynthesis due to cloudy periods)."

**Response**: We are delighted that the reviewer recognises that this discussion on signal noise is interesting and correct. As far as we know, no studies have been found in analysing the noise of NEE in such details. First, it is not easy or trivial to find a "highly smoothness estimator" while maintaining the main variations of signals. For example, replacing the gaps with a single value of the mean value is the smoothest way, but the temporal variations will be totally lost. In fact, IIP, as a highly non-linear estimator, has the ability to "learn" the pattern of signal and then implement the reconstruction. The problem raised in the reviewer's previous comment (and our response) is that *we would not be able to distinguish the gap-filling performance in artificial gaps even with a highly smooth estimation*. Based on these results, we speculate that random process embedded in the NEE signal ultimately determine the gap-filling performance – a fair explanation for these observations. Secondly, the de-noise method based on Fourier Transform is justifiable. We have showed that the full signal can be decomposed into two parts, one of which is a "close to a zero" mean superposed with a symmetric (e.g. Gaussian) distribution (i.e. features typical of noise). IIP can reconstruct the de-noised signal quite well and one should notice that the de-noised signal conserves the main information of temporal variations (see Fig. 5). In fact, all gap-filling methods are technically smoothness estimators and work by modelling patterns (structures) in the signal pattern (e.g. by drawing a single curve through a scatter of points in regression or by averaging (Mean) Diurnal Variation), to maintain the main trends while discarding fluctuations. As we have pointed out in the Discussion, the accuracy of IIP as a completely unsupervised process for filling data gaps, particularly when coupled with a de-noising algorithm, may contribute to bringing into focus underlying ecological

and meteorological mechanisms not identifiable a priori.

3) **Reviewer's comments**: "Some of the results interpretation are subjective and not justified. For example the fact that the reconstruction of the larger gap in summer in the DE-Gri site (figure 3) by the IIP is something to be positively evaluated because smoother and less noisy respect to MDS (page 5) needs to be proved. It is possible that the correct reconstruction is the one from the MDS method. . . only artificial (long) gaps can say which method is more close to the original measured data."

**Response**: We have made major corrections to the interpretation of result 3.1. We agree that only through using artificial gaps can we say whether either method is better. In our studies, little difference was found between methods.

To conclude, 1) IIP, despite being compact and completely unsupervised, shows a gap-filling performance at a same level of MDS when applied to the original NEE signal; 2) IIP outperforms MDS on de-noised NEE data. Either way, IIP is a robust gap-filling method for any structured data, including NEE.

[revised manuscript text omitted]
 outcomes between the two methods shown on the finger-prints plots, however, are not sufficient to determine their performance quantitatively. The difference was further evaluated for all twelve datasets at the data points of artificial gaps where the true NEE values were available. The gap filling error was simply calculated by taking the difference of the estimated and real values at those data points. The error distributions represented by the error bar plot (Fig. 4) showed that there was little difference between methods, i.e. comparable means and variances in the gap-filling error, even though we see different levels of smoothness from the contour plots. Mean values close to zero suggests that both methods provided nearly unbiased estimations for the NEE signal. Combining all twelve datasets categorized by gap types and using a single metric for errors (i.e. *RMSE*), we again found that the two methods were hardly distinguishable from each other (Table 2). As might have been expected, gap-filling error tended to increase as gap length increased for both methods. One should notice, however, the increase amount was relatively small, with a difference of ~1.2 in *RMSE* between the random and 14-day gap types (i.e. the two extremes). The facts above implied that, 1) Smoothly filling the gaps (by IIP) did not necessarily performed less well in terms of the estimation accuracy; 2) Large gaps did not significantly affect the gap-filling performance for either method. In fact, this result is consistent with a previous study (Falge et al., 2001) where the gap-filling residuals were not distinguishable by ANOVA. Both implications seem counter-intuitive, however, the simple way of understanding this is to recognise that both gap-filling methods "failed" to recover the missing signals. This happens if a signal contains a significant amount of noise/randomness which would be impossible for any method to recover.

In addition to the observation above, in Fig 4, the gap-filling error showed the most variation at site ITRo3 2013, while it had the least variation at site UKAMo 2010 irrespective of gap types. This raises the question of where do 
[revised manuscript text omitted]

| RMSE | 3.15 | 3.19 | 4.10 | 3.81 | 3.67 | 3.51 | 4.38 | 4.51 |

---

## Referee Comment (RC2) · Anonymous Referee #2 · 1 Oct 2016

The manuscript describes the concept of an image inpainting (IIP) method and its application to data gap filling for carbon dioxide fluxes (i.e., net ecosystem exchange; NEE). Since, as far as I know, flux researchers are not familiar with the IIP method and/or similar image analyses. It can provide important information and insight for the study area. While the manuscript is worth to be published, I have a critical question and some comments/questions on it.

A range of NEE should be different at site by site. In fact, the range of the site DEGri 2012 is from -30 to + 20 (Figure 3), and its of the site UKAMo 2010 is from -10 to +5 (Figure 6). Even the ranges are different, color scales at the both site are the same; the lowest is blue, around zero is green, and the highest is yellow. When those ranges

Interactive
comment

and scales are used for converting to gray scales, an representative NEE value of each gray scale is changeable at site by site, and it may lead to complex interpretations of "noise" because it is obtained from the difference between the gap filling values (i.e., scale of gray) and the real values. For an uniform analysis over the sites, why the fixed range for color/gray scales (for example, -30 to +20 for all site) is not applied? If there is some reason to use the specific range for color/gray scale for each site, please mention about it.

Other minor comments are listed below.

Title: It is hard to infer the contents of the manuscript by reading the term "Cahn-Hilliard inpainting". Please try to express this in different words.

Page 3, line 26-27: Can colored fingerprint figures (e.g., Figure 3 (b), (e), (h) and (k)) be applied to the IIP? Why are the figures converted to gray scale images? (Is the IIP limited only to the gray scale images?)

Page 4, line 18: What does the "T" mean in equation 1? There is no explanation.

Page 5, line 28: "patitioned" -> "partitioned" ?

Page 6, line 14-16 and Page 14, Figure 5(d): Richardson et al. (2006) showed that random errors of fluxes follow a double exponential distribution. Though the noise in the manuscript and the random errors in Richardson et al. (2006) are different concepts, how about also trying to fit the noise to the double exponential distribution?

Page 8, line 9-10: The authors noticed that the IIP performed less well for long gaps which gives rise to a question. Are the "long gaps" mentioned in the context a gap percentage of total data, or an absolute gap length? If a site has a one month data gap in a one year data set, the gap percentage is ca. 8% and, in my understanding, it may lead to a less accurate result. However, in the case of a one month data gap in a ten years data set using one fingerprint figure, the absolute gap length is the same but the gap percentage changes to small (ca. 0.8%), and will it produce a different result or the

same? It should be made clear whether the "long gaps" referred to are the percentage or the absolute length.

Page 18, Table 2: What does the "sample size" mean? Are the data points used for the IIP?

Reference: Richardson et al. (2006) A multi-site analysis of random error in tower-based measurements of carbon and energy fluxes, Agricultural and Forest Meteorology, 136, pp. 1–18.

---

## Author Comment (AC2) · 7 Oct 2016

We are very glad that the reviewer found this study innovative and important. We very much appreciate his/her constructive comments, which are penetrating and inspired, and have lead to some substantial improvements on this work. Building upon the revised manuscript (our response to reviewer #1), we have made further changes to the paper to incorporate the idea/issues raised by the current reviewer. The points are listed as follows and please refer to an updated manuscript attached in the supplementary PDF file.

1. About the colour scales of the fingerprint figures

**Reviewer's comment:**
"A range of NEE should be different at site by site. In fact, the range of the site DEGri 2012 is from -30 to + 20 (Figure 3), and its of the site UKAMo 2010 is from -10 to +5 (Figure 6). Even the ranges are different, color scales at the both site are the same; the lowest is blue, around zero is green, and the highest is yellow. When those ranges and scales are used for converting to gray scales, an representative NEE value of each gray scale is changeable at site by site, and it may lead to complex interpretations of "noise" because it is obtained from the difference between the gap filling values (i.e., scale of gray) and the real values. For an uniform analysis over the sites, why the fixed range for color/gray scales (for example, -30 to +20 for all site) is not applied? If there is some reason to use the specific range for color/gray scale for each site, please mention about it."

**Our response:**
We understand the confusion caused by the non-uniform colour scales and we did think about a consistent colour scale for all fingerprint figures, however this would reduce the intuitive understanding of these graphics (one per site), each of which has a distinct range of values. For example, if we adopted a uniform value range from $-30$ to $+20$, the contour plots for site UKAMo2010 would tend to be single-coloured and thus the flux variations represented by colours will become less distinguishable (see Fig. 1b below). Since we are not comparing the fingerprint figures between sites, the colour scales are better treated independently (e.g. colour yellow for one site has nothing to with the yellow for another).

As pointed out by the reviewer, the critical problem that may affect the IIP performance lies in the conversion of NEE value to grayscale (i.e. normalize NEE to (0,1)), which generates different NEE grayscale representations for different scale ranges. Originally, we used the following equation to convert/normalize the NEE values (Note: we have also included this in the manuscript at Page 3 line 25-30 and Page 4 line 1-4):

$$NEE_{gray} = \frac{NEE - NEE_{min}}{NEE_{max} - NEE_{min}}$$

where $NEE_{min}$ and $NEE_{max}$ are the minimum and maximum value of NEE respectively *for a given site*. If a universe grayscale were adopted, it means that a constant $NEE_{min}$ and $NEE_{max}$ would be used *for all sites*, which clearly would generate different values of $NEE_{gray}$. The fundamental question, however, is whether this transformation of NEE would have any impact on the gap-filling results. It turns out that this normalization method does not affect the IIP performance at all (see Fig. 2 below), suggesting that a simple linear transformation of NEE like this does not affect the computation output. Intuitively, we may say that IIP "senses" the relative gradient, irrespective of the actual value scale.

This result also supports our previous point that a unified colour scale is neither desirable, nor necessary because 1) it's not visually intuitive; 2) it does not affect the simulation output at all. Moreover, the colour information shown on the contour plot is, in fact, arbitrary, and for the purpose of illustration only. For example, we could group deep blue and light blue into a single colour that would not be distinguishable by eye, but the computer retains all the value variation within that single colour. In short, value changes continuously by given time step (e.g. half-hourly here), whilst the colour is discretised into sectors to provide a more intuitive view.

The reviewer also made another comment related to the colour scale.

**Reviewer's comment:**
"Page 3, line 26-27: Can colored fingerprint figures (e.g., Figure 3 (b), (e), (h) and (k)) be applied to the IIP? Why are the figures converted to gray scale images? (Is the IIP limited only to the gray scale images?)"

**Our response:**
There are three reasons why we limit this IIP algorithm to grayscale images: 1) This inpainting algorithm was originally developed for grayscale images; 2) NEE is a one-dimensional vector and a projection of this vector onto an RGB value (3-d vector) does not add extra information; 3) Most importantly, as discussed above, we would lose information if discretised colours were used to represent NEE values because

the variation within colour (hue) is neglected. The half-hourly variation is already the finest resolution that can be "resolved" by the differential equation of this IIP (i.e. the differential equation represents function's change/gradient).

Nevertheless, this does point out a potential follow-up study on an extension of IIP to higher dimensional gap-filling. We have added the following arguments in the discussion section at Page 9, line 3-13:

"We limit the use of IIP to grayscale images because 1) This inpainting algorithm was originally developed for grayscale images [Burger et al., 2009]; 2) The NEE is a one-dimensional vector and a projection of this vector onto an RGB value (3-d vector) does not add extra information; 3) Most importantly, we would lose information if discretised colours were used to represent NEE values because the variation within colour (hue) would be neglected. In fact, the half-hourly variation is already the finest resolution that can be "resolved" by the differential equation in this IIP (i.e. the differential equation represents that function's change/gradient). Nevertheless, a recent study has extended the application of the Cahn-Hilliard IIP to colour images [Cherfils et al., 2016] and other techniques for colour image inpainting have been proposed (e.g. [Bertalmio et al., 2000]). These developments would allow us to apply IIP-based gap-filling to higher dimensions by incorporating constraining "information" derived from other fluxes or other environmental variables. Thus, potential follow-up studies could focus on improving representation of NEE in colour images, using, for example, Green for NEE, Blue for latent heat flux and Red for sensible heat flux."

2.Responses to reviewer's other comments
**Reviewer's comment:**
"Title: It is hard to infer the contents of the manuscript by reading the term "Cahn-Hilliard inpainting". Please try to express this in different words."
**Our response:**
We agree with the reviewer about the title and we now propose an alternative as: "A robust gap-filling method for Net Ecosystem CO2 Exchange based on image inpainting

(IIP)"

**Reviewer's comment:**
"Page 4, line 18: What does the "T" mean in equation 1? There is no explanation."
**Our response:**
This part indeed need a clarification. We have added the following description at Page 4 Line 29-31:
"Where $a$ is the frequency mode in Fourier domain. Since $a$ is a complex number, $|a|$ is the complex modulus. T is the threshold chosen to filter the frequency according to its energy (i.e. the modulus of a given frequency). One may draw a histogram of all frequency moduli (i.e. power spectral density) to help choosing a threshold."

**Reviewer's comment:**
"Page 6, line 14-16 and Page 14, Figure 5(d): Richardson et al. (2006) showed that random errors of fluxes follow a double exponential distribution. Though the noise in the manuscript and the random errors in Richardson et al. (2006) are different concepts, how about also trying to fit the noise to the double exponential distribution?"
**Our response:**
This is certainly a very interesting point raised by the reviewer and thanks to his insightful suggestion, we can further highlight the robustness of our Fourier transform-based de-noise method. The following changes have been made to manuscript:

- Fig. 5 in the original manuscript has been separated into three individual graphs as Fig. 5, Fig. 6 and Fig. 7 in this revised version, representing the de-noised temperature, de-noised NEE and the residual distributions respectively.

- The double exponential distribution was fitted to the NEE residual distribution, which was plotted with the normal fit in Fig. 7b. It was indeed a better fit than the normal distribution.

- Accordingly, the statements in the result section have been modified at Page 6 Line 20-33 and Page 7 Line 1-11.

**Reviewer's comment:**
"Page 8, line 9-10: The authors noticed that the IIP performed less well for long gaps which gives rise to a question. Are the "long gaps" mentioned in the context a gap percentage of total data, or an absolute gap length? If a site has a one month data gap in a one year data set, the gap percentage is ca. 8% and, in my understanding, it may lead to a less accurate result. However, in the case of a one month data gap in a ten years data set using one fingerprint figure, the absolute gap length is the same but the gap percentage changes to small (ca. 0.8%), and will it produce a different result or the same? It should be made clear whether the "long gaps" referred to are the percentage or the absolute length."
**Our response:**
As mentioned in the discussion section of the original manuscript, it is the "diameter" of gaps that determines the IIP performance. In fact, we have validated this by adding up to 30% random artificial gaps (of small diameter) at site UKAMo2010. Results (omitted here) showed that there was no significant difference between the case with 10% and 30% random gaps, suggesting that the IIP performance was not much affected as long as the holes in the image are small. Intuitively, this is quite understandable: the information missing from small gaps is not critical because the surrounding non-gaps still contain enough information for IIP to propagate. To answer the reviewer's question, in the context of this paper, "long gaps" means their absolute length. We agree on a clarification of the statement and the relevant sentence in the manuscript has been slightly modified as:
"Firstly, IIP performed less well for long gaps where the information density is low (i.e. the diameter of gaps or the absolute gap length is large)."

**Reviewer's comment:**

"Page 18, Table 2: What does the "sample size" mean? Are the data points used for the IIP?"

**Our response:**

Sample size in Table 2 means the total number of the "valid" artificial gaps for 12 datasets. We combine all 12 datasets for each gap type to show the RMSE of gap-filling estimation (i.e. overall performance of IIP and MDS). We have changed the term "sample size" to "Total number of valid artificial gaps" in Table 2.

The table caption has been modified as:

"Table 2 Overall gap-filling error of all 12 datasets estimated for each artificial gap type. "Valid" artificial gaps are the artificial gaps that do not overlap with real gaps."

Also, two sentences were added to Line 8-10 at Page 4:

"It should be noted that any artificial gaps that overlapped with real gaps were not included in the calculation because there were no known values at those positions. On average, more than 1000 artificial gaps were valid for each gap type at each site."

Reference:

Bertalmio, M., Sapiro, G., Caselles, V., Ballester, C., 2000. Image inpainting. Proc. 27th Annu. Conf. Comput. Graph. Interact. Tech. SIGGRAPH 00 2, 417–424. doi:10.1145/344779.344972

Burger, M., He, L., Schönlieb, C.-B., 2009. Cahn–Hilliard Inpainting and a Generalization for Grayvalue Images. SIAM J. Imaging Sci. 2, 1129–1167. doi:10.1137/080728548

Cherfils, L., Fakih, H., Miranville, A., 2016. A Cahn–Hilliard System with a Fidelity Term for Color Image Inpainting. J. Math. Imaging Vis. 54, 117–131. doi:10.1007/s10851-015-0593-9

Please also note the supplement to this comment:
http://www.geosci-model-dev-discuss.net/gmd-2016-108/gmd-2016-108-AC2-supplement.pdf

[Figure]

**Fig. 1.** Original colour scale (a) ranging from -10 to +5, compared with the scale ranging from -30 to +20 (b) for NEE data at site UKAMo_2010

[Figure]

**Fig. 2.** IIP performance for NEE normalized per individual site (Top) and by all sites (Bottom)

**Supplement:**

[revised manuscript text omitted]

$$NEE_{gray} = \frac{NEE - NEE_{min}}{NEE_{max} - NEE_{min}}, \qquad\qquad (1)$$

Where $NEE$ is the original data. $NEE_{min}$ and $NEE_{max}$ are the minimum and maximum values of $NEE$ respectively for a given site. $NEE_{gray}$ is the normalized $NEE$ with the value range (0, 1). The gaps are then filled by the inpainting algorithm based on the code from the MATLAB Central File Exchange (Schönlieb, 2011). Finally, the filled grayscale $NEE$ is converted back to the original by using the inverse of Eq. (1).

5  In order to evaluate the performance of the gap-filling methods on the data points where real values exist, short and long artificial gaps amounting to about 10% of each dataset are considered in the simulations. Concretely, for the short type, half-hourly gaps are added uniformly randomly to the original NEE signal, while gaps with length of 3-day, 7-day and 14-day are added respectively (Fig. 2). It should be noted that any artificial gaps that overlapped with the real gaps were not included in the calculation because there were no known values at those positions. On average, more than 1000 artificial gaps were valid
10  for each gap type at each site.

**2.3 Noise reduction**

To start with, we need to clarify what "noise" means in the context here. For a given signal, it can be partitioned into two parts: the trend part and the stochastic part. The trend part is called the de-noised signal and the stochastic part is referred as the noise. The noise characterized the randomness of a signal. As there is no general rule for reducing the noise from a NEE
15  signal, the following assumptions are made for validating a de-noise method:

1.    Noise has zero mean and symmetric/unbiased distribution;

2.    Covariance between the noise and the de-noised signal is negligible (close to zero);

3.    The difference between the before and after de-noising are small in the cumulative temperature and NEE.

The point 1 and 2 are used to show that the noise part was similar to a stochastic, unstructured and non-correlated signal.
20  Since the underlying pattern of NEE is unknown, the cumulative and average temperature and NEE are used to show that the important information still remains after the de-noising process (see details in Results).

A simple method based on the Fourier transform of an entire time-series is used to reduce noise in the NEE and temperature signals. This process is illustrated by the block diagram:

$$x(n) \rightarrow \boxed{\mathcal{F}\big(x(n)\big)} \rightarrow \boxed{Threshold: \hat{x}(k) = g \cdot \mathcal{F}(x)} \rightarrow \boxed{\mathcal{F}^{-1}\big(\hat{x}(k)\big)} \rightarrow y(n)$$

25  where x($n$) is the original "noisy" signal in the time domain ($n$), with any gaps initialized with the mean value of the rest of the signal. $\mathcal{F}$ and $\mathcal{F}^{-1}$ are the fast Fourier transform and its inverse respectively. $\hat{x}(k)$ is the filtered signal at frequency $k$ and $y(n)$ stands for the de-noised signal. The threshold step was carried out using a simple binary function:

$$g(a) = \begin{cases} 0, & |a| \leq T \\ 1, & |a| > T \end{cases}, \tag{2}$$

Where $a$ is the frequency mode in Fourier domain. Since $a$ is a complex number, $|a|$ is the complex modulus. T is the
30  threshold chosen to filter the frequency according to its energy (i.e. the modulus of a given frequency). One may draw a histogram of all frequency moduli (i.e. power spectral density) to help choosing a threshold.

Two more sophisticated noise reduction techniques, the short-time Fourier transform and wavelets (each using various sized windows) were also tested in our study, but did not show distinct advantages over the simple Fourier transform, and the results are not presented here.

A dimensionless quantity is used to measure how much noise has been removed by the de-noising process. In image processing, the quality of a signal can be expressed quantitatively as the signal-to-noise ratio (SNR) (Schowengerdt, 2006), denoted as:

$$SNR = \frac{\sigma_{signal}}{\sigma_{noise}} ,$$ (3)

where $\sigma_{signal}$ and $\sigma_{noise}$ are the standard deviation of post-filter signal and the standard deviation of the filter-out signal (noise) amplitude in the Fourier domain, respectively.

**2.4 Analysis**

The two gap-filling methods were applied to the original and the noise-reduced NEE data from 12 years of measurements at 6 European sites respectively. Following Moffat et al. (2007), we assumed that the differences between the traditional methods are negligible, and therefore comparisons were only conducted between IIP and MDS. Initially, we applied the two methods to the original NEE datasets and measure their performance on four types of artificial gaps, including short random gaps and long gaps up to 14 days. Further simulations were then conducted to show how noise or random structures in the signal may affect the gap-filling performance by partitioning the original signal using Fourier transform.

**3 Results**

**3.1 Gap filling the NEE data with artificial gaps**

Figure 3 shows an example of the comparison of the gap-filling performance between IIP and MDS on the post-QC NEE data with artificial gaps at site DEGri 2012. The general temporal patterns revealed by the two gap-filling methods are very similar across all four gap types. Clear diurnal and seasonal variations were well captured by both methods. In contrast to MDS, contour structures and boundaries generated by IIP are smoother or less "noisy". This can be seen mostly clearly from a large gap (~ 2 weeks) in the middle of the year. Effects of gap type on the gap-filling performance were minimum and can hardly be noticed from the contour plots, which may suggest that IIP was able to reconstruct the signal even with the occurrence of the long-gap type up to 14 days.

The difference of gap-filling outcomes between the two methods shown on the finger-prints plots, however, are not sufficient to determine their performance quantitatively. The difference was further evaluated for all twelve datasets at the data points of artificial gaps where the true NEE values were available. The gap filling error was simply calculated by taking the difference of the estimated and real values at those data points. The error distributions represented by the error bar plot (Fig. 4) showed that there was little difference between methods, i.e. comparable means and variances in the gap-filling error,

even though we see different levels of smoothness from the contour plots. Mean values close to zero suggests that both methods provided nearly unbiased estimations for the NEE signal. Combining all twelve datasets categorized by gap types and using a single metric for errors (i.e. *RMSE*), we again found that the two methods were hardly distinguishable from each other (Table 2). As might have been expected, gap-filling error tended to increase as gap length increased for both methods.

5    The increase amount, however, was relatively small, with a difference of ~1.2 in *RMSE* between the random and 14-day gap types (i.e. the two extremes). The above results implied that, 1) Smoothly filling the gaps (by IIP) did not perform less well in terms of the estimation accuracy; 2) Large gaps did not significantly affect the gap-filling performance for either method. This result is in fact consistent with a previous study (Falge et al., 2001) where the gap-filling residuals were found to be not distinguishable by ANOVA. Both implications seem counter-intuitive, however, the simple way of understanding this is to

10    recognise that both gap-filling methods "failed" to recover the missing signals. This happens if a signal contains a significant amount of noise/randomness which would be impossible for any method to recover.

In addition to the observation above, in Fig. 4 the gap-filling error showed the most variation at site ITRo3 2013, while it had the least variation at site UKAMo 2010 irrespective of gap types. This raises the question of where does the variation among datasets originates from? In other words, why was the estimation from some dataset always better than others? The

15    estimation confidence represented by one standard deviation (i.e. the span of the error bar) should be nearly zero for an ideally clean, noise-free image. As noise increases, the span representing the estimation uncertainty becomes wider. We will address this problem in the next part by demonstrating that the variation of the gap-filling error originates from some random structures in the signal.

**3.2 Random structures/noise affect the gap-filling performance**

20    To start with, the temperature and the NEE signals were partitioned into two components respectively. To check that the de-noising procedure did not introduce bias, the average and cumulative temperature and NEE were compared before and after the signal de-noising process. Examples of the de-noising process are shown in Fig. 5&6 for temperature and NEE respectively at site UKAMo in 2010. The average and cumulative temperature signals are almost identical (Fig. 5a), suggesting that the system energetics remain the same even though some structures of randomness has been removed from

25    the time series of temperature. Furthermore, the distribution of the removed part of temperature or simply the noise distribution shows a good agreement with the normal distribution (Fig. 7a), implying a Gaussian-structured noise embedded in the original temperature signal. Fig. 5b & 5d show the fingerprint plots for the raw and de-noised temperature signals respectively. It is clear that the de-noising process smoothed out some variations from the raw signal, resulting in a much cleaner image.

30    A similar result can be found in the de-noised NEE (Fig. 6) even though the SNR (~1.2) is much lower than that of temperature, suggesting that the NEE was initially noisier than the temperature. The value of SNR was determined by the thresholding step (Eq. 2) and for de-noising the NEE signal, 1.2 of SNR was found to be approximately a lower limit of the noise removal in order to maintain a clear diurnal and seasonal variation (Fig. 6d). We show this largely smoothed NEE to

demonstrate that the average and cumulative NEE after the de-noising are still good approximates to the original ones. For any less smoothed NEE with higher SNR values the cumulative NEE fits even better. Unrealistic fluctuations of the original NEE appear mostly at night-time and the de-noising method seems to fix this, as a traditional regression method would work, by replacing the night-time NEE with some simple variations (Fig. 6a), which might be the main cause for the

5    discrepancy in the accumulative de-noised NEE from the original (Fig. 6c). Intuitively this abnormality in the NEE at night-time supports our speculation that noise exists in the NEE signal, which would affect the gap-filling performance by introducing random error variations. The distribution of the noise part of NEE, however, is not a good fit to the normal distribution but is steeper and more like the double exponential distribution (Fig. 7b) as suggested by a previous study on the random error structure in NEE (Richardson et al., 2006). This may further suggest that the type of noise embedded in the

10   NEE signal is, as might be expected, more complicated and different from the normally-distributed noise found in the functionally less complex temperature flux fingerprint.

[revised manuscript text omitted]

We limit the use of IIP to grayscale images because 1) This inpainting algorithm was originally developed and justified for grayscale images (Burger et al., 2009); 2) The NEE is a one-dimensional vector and a projection of this vector onto an RGB value (3-d vector) does not really add extra information; 3) Most importantly, we would lose information if discretised colours were used to represent NEE values because the variation within colour (hue) would be neglected. In fact, the half-hourly variation is already the finest resolution that can be "resolved" by the differential equation in this IIP (i.e. the differential equation represents function's change/gradient). Nevertheless, a recent study has extended the application of the Cahn-Hilliard IIP to colour images (Cherfils et al., 2016) and other techniques for colour image inpainting have been proposed (e.g. (Bertalmio et al., 2000)). These developments would allow us to apply the IIP-based gap-filling to higher dimensions by incorporating constraining "information" derived from other fluxes or other environmental variables. Thus, potential follow-up studies could focus on improving representation of NEE in colour images, using, for example, Green for NEE, Blue for latent heat flux and Red for sensible heat flux.

[revised manuscript text omitted]

5  **Figure 2 Four gap types generated are (a) random gaps; (b) 3-day; (c) 7-day; (d) 14-day. White strips represent the gap positions. Number of gaps for each gap type was about 10% of the whole year (i.e. ~1752 data points).**

[Figure]

**Figure 3. Image inpainting vs Marginal Distribution Sampling for the four gap types at site DEGri 2012. The middle column (i.e. (b), (e), (h) and (k)) are the original NEE data with random, 3-day, 7-day and 14-day artificial gaps respectively. The left (i.e. (a), (d), (g) and (j)) and right (i.e. (c), (f), (i) and (l)) column represent the gap-filling results from IIP and MDS accordingly.**

[Figure]

**Figure 4. Comparisons of the gap-filling error between IIP and MDS for the four gap types, i.e. (a) Random gaps; (b)3-day gaps; (c) 7-day gaps; (d) 14-day gaps. Unit of the error values are in $\mu mol\ m^{-2}\ s^{-1}$. Error bar plot with mean ± one standard deviation of the absolute errors for each dataset.**

[Figure]

**Figure 5. An example of a highly de-noised temperature data at site UKAMo for year 2010. Raw and de-noised half-hourly (a) and cumulative (c) temperature. 2-d visualizations (fingerprints) of raw and de-noised temperature in (b) and (d) respectively.**

[Figure]

**Figure 6. An example of a highly de-noised NEE data at site UKAMo for year 2010. Raw and de-noised half-hourly (a) and cumulative (c) NEE. 2-d visualizations (fingerprints) of raw and de-noised NEE in (b) and (d) respectively.**

[Figure]

**Figure 7. Noise distribution of temperature (a) and NEE (b). NEE was fitted to both the normal distribution (red line) and the double exponential distribution (green line).**

[Figure]

**Figure 8. Data from UKAMo_2010. De-noising the NEE dataset and its gap filling. NEE ft is the de-noised NEE by Fourier transform. NEE AGaps stands for the post-denoise NEE with artificial gaps and real gaps. SNR levels are (a) 2.34, (b) 1.68, (c) 1.30, (d) 1.17.**

[Figure]

**Figure 9 Site: ITRo3_2013.   De-nosing the NEE dataset and its gap filling. NEE ft is the de-noised NEE by Fourier transform. NEE AGaps stands for the post-denoise NEE with artificial gaps and real gaps. SNR levels are (a) 24.87, (b) 1.90, (c) 1.50, (d) 1.35.**

[Figure]

**Figure 10. Gap-filling errors response to the ratio of energy remaining after de-noising (SNR). (a) Site: UKAMo_2010; (b) Site: ITRo3_2013. The gap-filling error simply refers to the difference between the gap-filling values and the real value at artificial gaps. Error bar plot stands for the mean ± one standard deviation of the errors.**

5   **Table 1. Description of the datasets from the European fluxes database. The gap percentage was calculated by counting the half-hourly data missing of a whole year (i.e. 17520 of data records for 365 days).**

| Site-ID | Location | Vegetation Type | Lat/Long | Year | Gap(%) | Post-QC Gap(%) | PI |
|---------|----------|-----------------|----------|------|--------|----------------|-----|
| **UKAMo** | Scotland | Peatland | -3.23°, 55.79° | 2010 | 9.9% | 30.2% | Marc Sutton |
| **UKEBu** | Scotland | Grassland | -3.20°, 55.86° | 2010 | 26.4% | 44.8% | Marc Sutton |
| **DEGeb** | Germany | Cropland | 10.91°, 51.10° | 2010 | 43.1% | 55.9% | Olaf Kolle, Mathias Herbst |
| **DEGri** | Germany | Grassland | 13.51°, 50.95° | 2010 | 9.9% | 31.1% | Christian Bernhofer |
| | | | | 2011 | 17.9% | 37.2% | |
| | | | | 2012 | 11.0% | 29.5% | |
| **ITRo3** | Italy | Cropland | 11.92°, 42.38° | 2011 | 23.6% | 50.1% | Dario Papale |
| | | | | 2012 | 22.0% | 48.4% | |
| | | | | 2013 | 7.1% | 42.7% | |

| ITRo4 | Italy | Savannah | 11.92°, 42.37° | 2011 | 27.7% | 56.7% | Dario Papale |
| | | | | 2012 | 19.1% | 47.8% | |
| | | | | 2013 | 23.4% | 52.2% | |

**Table 2 Overall gap-filling error of all 12 datasets estimated for each artificial gap type. "Valid" artificial gaps are the artificial gaps that do not overlap with real gaps.**

| Gap types | Random | | 3-Day | | 7-Day | | 14-Day | |
|---|---|---|---|---|---|---|---|---|
| Gap-filling methods | IIP | MDS | IIP | MDS | IIP | MDS | IIP | MDS |
| Total number of valid artificial gaps | 11817 | 11817 | 11493 | 11493 | 10593 | 10593 | 13416 | 13416 |
| RMSE | 3.15 | 3.19 | 4.10 | 3.81 | 3.67 | 3.51 | 4.38 | 4.51 |